# Miocene restriction of the Pacific-North Atlantic throughflow strengthened Atlantic overturning circulation

Valeriia Kirillova[1], Anne H. Osborne [1], Tjördis Störling [1] & Martin Frank [1]

Export of warm and salty waters from the Caribbean to the North Atlantic is an essential component of the Atlantic Meridional Overturning Circulation (AMOC). However, there was also an active AMOC during the Miocene, despite evidence for an open Central American Seaway (CAS) that would have allowed low-salinity Pacific waters to enter the Caribbean. To address this apparent contradiction and to constrain the timing of CAS closure we present the first continuous Nd isotope record of intermediate waters in the Florida Strait over the past 12.5 million years. Our results indicate that there was no direct intermediate water mass export from the Caribbean to the Florida Strait between 11.5 and 9.5 Ma, at the same time as a strengthened AMOC. After 9 Ma a strong AMOC was maintained due to a major step in CAS closure and the consequent cessation of low-salinity Pacific waters entering the Caribbean.

---

[1] GEOMAR Helmholtz Centre for Ocean Research Kiel, Kiel, Germany. Correspondence and requests for materials should be addressed to A.H.O. (email: aosborne@geomar.de)

Of the $30 \pm 1.5$ Sverdrup (Sv, $1 \times 10^6 \, m^3 \, s^{-1}$) of water transported via the Florida Current today, 13 Sv continue to flow northwards as the Gulf Stream[1], thus constituting a major contribution to the Atlantic Meridional Overturning Circulation (AMOC), which has a mean deep volume transport of $18 \pm 2.5$ Sv at 24°N[2]. Most model experiments suggest that an open Central American Seaway (CAS) allowed low-salinity waters from the equatorial Pacific to enter the Atlantic at mid-latitudes resulting in an overall weaker AMOC (e.g.[3–6]), and a possible link between CAS closure and changes in North Atlantic Deep Water production has been widely discussed (e.g.[7–10]). The timing of CAS closure is thus of crucial importance but available reconstructions are controversial, with much of the controversy hinging on the definition of closure[11]. On the basis of geochemical and geochronological data and paleomagnetic reconstructions, recent studies have argued that the CAS essentially disappeared completely as early as 15 Ma[12–14], although allowing that transient and shallow seaways may have persisted until 10 Ma[14]. This, however, contradicts biostratigraphic evidence for a gradual shoaling from 15 Ma onwards, with the seaway still 1000-m deep at 12 Ma[15]. The appearance of a significant and permanent surface water salinity difference between the eastern equatorial Pacific and the Caribbean after ~4.7 to 4 Ma[16–20] points towards a later CAS closure, although changes in sea-surface properties could alternatively be explained by a switch from predominantly El Niño-like to predominantly La Niña-like conditions[21,22]. What constitutes 'closure' is also under debate where the fossil record is concerned. If shallow seaways with sufficiently high currents, similar to the Indonesian throughflow[23], were present in the Pliocene CAS, this may have prevented migration of species between North and South America[20] and could explain the divergence of marine molecular DNA on either side of the CAS after 3 Ma[20] and the fossil evidence for the Great American Biotic Interchange[24] from 2.6 Ma onwards[25]. If the currents in the CAS were not strong, or if the seaways were transient, then this could

explain why a recent study combining fossil and molecular data found waves of terrestrial organism dispersal at ~20 and 6 Ma and divergence of Atlantic and Pacific marine organisms at ~23 and ~7 Ma[26].

In the light of the ambiguous evidence for the timing of CAS closure, it is pertinent to examine the progressive restriction of water mass exchange via the CAS in more detail and to compare these results with model predictions of the consequences of variations in Pacific water export to the Caribbean and the Atlantic. The radiogenic Nd isotope composition of past seawater has been used extensively to reconstruct changes in the mixing and provenance of water masses (cf. ref.[27]). The longest record of the Gulf Stream $\varepsilon_{Nd}$ evolution from 850 m water depth in the northwest Atlantic extends back to 8.5 Ma and shows a shift towards less radiogenic compositions between 8.5 and 5 Ma, which was interpreted as a reduction in the advection of intermediate-depth Pacific water to the western North Atlantic[28], a conclusion supported by model results[29]. For deep waters, sedimentary $\varepsilon_{Nd}$ records from the Caribbean extending back to 18 Ma show that the irreversible shift away from Pacific-like compositions already occurred earlier, at 10.7 Ma[10]. However, in apparent contrast to most model predictions (ref.[5] and references therein), episodes of increased Pacific through-flow in the Middle to Late Miocene Caribbean (between ~12.5 and 10 Ma) were shown to have coincided with increased AMOC strength[10].

In this contribution, we specifically compare the evolution of the export of intermediate-depth Pacific waters to the western North Atlantic to records of AMOC strength[30–32]. We present a continuous new 12.5 Myr seawater $\varepsilon_{Nd}$ record obtained from sedimentary foraminifera coatings of Ocean Drilling Program (ODP) Site 1006 (658 m water depth) located on the leeward slope of the Great Bahama Bank (GBB) (Fig. 1), which together with Site 1000 in the central Caribbean (916 m water depth) for the first time constrains the timing of CAS closure to intermediate-depth Pacific waters and also allows reconstruction

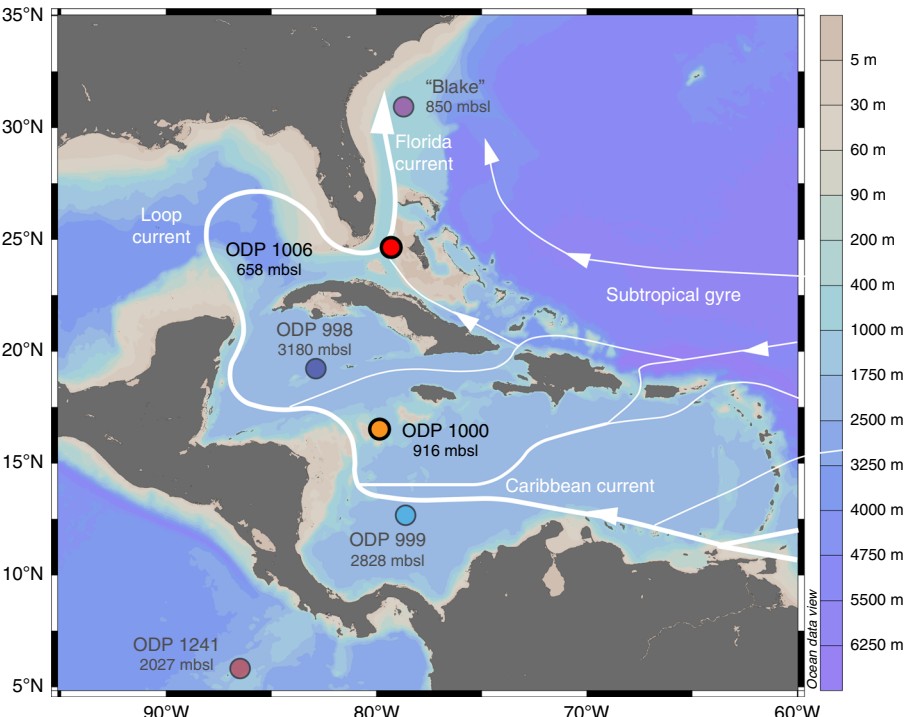

**Fig. 1** Study area and sample sites. Locations and water depths of Ocean Drilling Program (ODP) Sites from which the new data were obtained (1000 and 1006) and of other sites referred to in this study (ODP 998, 999 and 1241, Blake crust). The main ocean currents affecting the sites are shown in white[35, 36, 61]. Map produced using Ocean Data View[62]

of the origin of western North Atlantic (Gulf Stream) waters during the Miocene–Pliocene.

## Results

**Seawater and detrital Nd-isotope signatures.** Comparison of surface sediment and water samples from the vicinity of ODP Site 1006 confirms that uncleaned planktonic foraminifera reliably recorded the local bottom water $\varepsilon_{Nd}$ signal[33] (Supplementary Figs. 1 and 2). The oldest part of the new seawater record of Site 1006 reveals highly radiogenic $\varepsilon_{Nd(t)}$ values of up to −4.6 prior to 11.5 Ma dropping sharply to −8.1 at 10.8 Ma (Fig. 2). Seawater $\varepsilon_{Nd(t)}$ values then became slightly more radiogenic again between 9.5 and 8 Ma (−6.6 to 7.3 $\varepsilon_{Nd(t)}$) before decreasing to −8.8 $\varepsilon_{Nd(t)}$ by 6.8 Ma. Thereafter, the $\varepsilon_{Nd}$ signatures at Site 1006 fluctuated between −8.4 and −6.2 until 3.1 Ma to finally become less radiogenic, approaching core-top values of ~−9 to −10 from 0.85 Ma onwards. The $\varepsilon_{Nd(t)}$ signature of the detrital fraction at Site 1006, with the exception of a single sample at 7.5 Ma, varied between −5.6 and −8.5 from 12.5 to 4 Ma and became less radiogenic thereafter (−8.6 to −10.7 $\varepsilon_{Nd(t)}$) (Supplementary Fig. 3).

New Miocene data were also produced to extend the existing foraminiferal and detrital $\varepsilon_{Nd(t)}$ records of Site 1000[34] and now allow direct comparison of the long-term changes at the two intermediate-depth sites (Fig. 2, Supplementary Fig. 3). The early part of the Site 1000 seawater record is significantly more radiogenic than that of ODP 1006 with $\varepsilon_{Nd(t)}$ values reaching −2.6 to −3.1 between 13.6 and 12.1 Ma (Fig. 2). This is followed by an $\varepsilon_{Nd(t)}$ decrease to −7.8 between 12.1 and 8.4 Ma and a slight increase to −6.6 between 7.3 and 6.8 Ma. Unlike the Site 1006 record, the detrital fraction of Site 1000 sediments shows a distinct step-change from $\varepsilon_{Nd(t)}$ values between −2.2 and −7.6

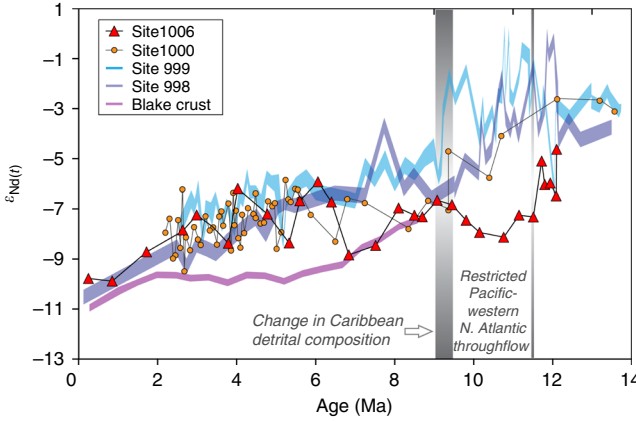

**Fig. 2** Seawater Nd isotope records. Seawater $\varepsilon_{Nd}$ records ($\varepsilon_{Nd} = [(^{143}Nd/^{144}Nd_{(sample)})/(^{143}Nd/^{144}Nd_{(CHUR)} −1] \times 10^4$, whereby CHUR stands for the Chondritic Uniform Reservoir ($^{143}Nd/^{144}Nd = 0.512638$)[63] of Ocean Drilling Program (ODP) Site 1006 (uncleaned foraminifera, this study), Site 1000 (uncleaned foraminifera, this study and ref. [34]), Sites 998 and 999 (fish teeth and uncleaned foraminifera[10, 34], and of ferromanganese crust Blake (BM 1963.897))[28]. Samples have been corrected to $\varepsilon_{Nd(t)}$ to take into account the ingrowth of $^{143}Nd$. All previously published data of the Blake crust and ODP Sites 998 and 999 are presented as thick lines that include the 2$\sigma$ external reproducibility of the analyses. For clarity, the error bars for the new data are excluded in this figure, but are shown in Supplementary Fig. 3. The Blake record only starts at 8.5 Ma. The vertical lines indicate the periods of time when Caribbean and Florida Strait seawater $\varepsilon_{Nd}$ signatures diverged between 11.5 and 9.5 Ma and converged again between 9.5 and 9 Ma. The detrital composition at ODP Site 1000 changed after 9 Ma (Supplementary Fig. 3)

prior to 9.4 Ma to between −8.3 and −10.5 after 8.4 Ma (Supplementary Fig. 3).

## Discussion

Prior to 12.1 Ma, seawater $\varepsilon_{Nd(t)}$ at Sites 1006 and 1000, as well as at deep Caribbean Sites 998 and 999[10], was within a range from −4.6 to −2, reflecting a strong Pacific throughflow into the western North Atlantic (Fig. 2). Pacific waters could have been transported to Site 1006 either via the Florida Strait or, as shown in model simulations[6], through entrainment in the Atlantic gyre, or a combination of both. Some contribution of radiogenic Nd derived from partial dissolution of volcanic ashes may be expected in this tectonically highly active region but a previous study[10] found no correlation between $\varepsilon_{Nd}$ signatures and volcanic ash mass accumulation rates at Caribbean Sites 999 and 998. An overall less radiogenic seawater $\varepsilon_{Nd(t)}$ signal at Site 1006 compared to Sites 998, 999 and 1000 is consistent with decreasing $\varepsilon_{Nd}$ signatures with increasing distance from the CAS, given that the Pacific signal was continuously mixed with Atlantic waters with a less radiogenic $\varepsilon_{Nd}$ signature[28,35,36]. Likewise, any particulate material from the Pacific transported across the CAS would more likely have been deposited in the vicinity of Site 1000 than at Site 1006, which explains why the detrital fraction at Site 1000 was on average ~2.5 $\varepsilon_{Nd}$ units more radiogenic than at Site 1006 prior to 9 Ma (Supplementary Fig. 3). Partial dissolution of volcanic particles along the path of the recirculated gyre waters or a marked change in the Southern Atlantic end-member composition would be further possible sources of radiogenic Nd, but there is no evidence to support this (see Supplementary Discussion).

After 12.1 Ma, seawater $\varepsilon_{Nd(t)}$ at Caribbean Site 1000 became progressively less radiogenic by more than 3 $\varepsilon_{Nd}$ units to reach −7.8 by 8.4 Ma. This decrease is similar to the overall trend at two deep Caribbean Sites, which together with increasing sediment accumulation rates was attributed to a decrease in the inflow of more radiogenic and corrosive Pacific waters entering the basin via the closing CAS[10]. Our new data show that the composition of the intermediate-depth Caribbean was strongly impacted by CAS closure in the Late Miocene, as reflected by a marked drop to less radiogenic detrital $\varepsilon_{Nd}$ signatures after 9 Ma. The $\varepsilon_{Nd(t)}$ and $^{87}Sr/^{86}Sr$ signatures of the detrital material (Fig. 3) suggest that the main source of the material delivered to Site 1000 between 14 and 9 Ma was the Mexican volcanic belt. After 9 Ma, the supply of

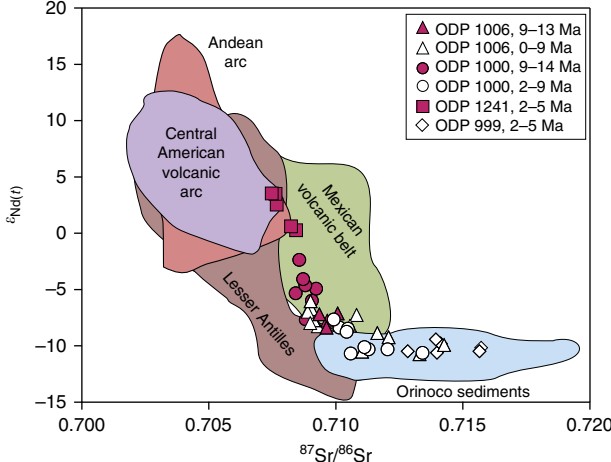

**Fig. 3** Detrital fraction Nd and Sr isotope compositions. Comparison of Nd and Sr isotope compositions of the detrital fraction of Ocean Drilling Program Sites 1006, 1000, 999 and eastern equatorial Pacific Site 1241 (this study and ref. [34]) and potential sources (data obtained via the GEOROC database http://georoc.mpch-mainz.gwdg.de/georoc/)

Orinoco sediments played the most important role, despite evidence that uplift and mountain building in the Northern Andes from ~12 Ma actually re-routed the drainage system in northern South America away from the Caribbean and towards the tropical Atlantic[37]. The shift in detrital material composition at Site 1000 therefore indicates a major change in the overall Caribbean subsurface circulation pattern.

We find that an initial decrease in seawater $\varepsilon_{Nd}$ at Site 1006 occurred earlier than at the Caribbean Sites and reached −8.1 at 10.8 Ma. This marked divergence from the Site 1000 record suggests that there was a change in the regional hydrography between ~11.5 and 9.5 Ma and that Atlantic waters with a less radiogenic Nd isotope composition dominated the signal at Site 1006. Continuous sedimentation of the Santaren Drift since 12.4 Ma[38] argues against tectonic uplift of a sill in close vicinity of Site 1006 being responsible for the divergence between the Caribbean and Florida Strait signals. The disconnection was more likely caused by a hydrographic barrier caused by the dominance of the circum-tropical current between the Pacific and Caribbean over the intra-Caribbean current connecting the Northern and Southern Caribbean, as was suggested on the basis of nannofossil assemblages[39]. Whatever the exact cause of the disconnection was, the evidence from the seawater $\varepsilon_{Nd}$ records clearly shows that there was no significant export of radiogenic intermediate-depth Pacific waters to the western North Atlantic during this time interval.

The convergence of the Caribbean and Florida Strait $\varepsilon_{Nd}$ records between 9.5 and 8 Ma documents the full hydrographic reconnection of the two intermediate-depth Sites (Fig. 2). ODP Site 1000 is located in the Pedro Channel, which crosses the Northern Nicaraguan Rise[40]. According to tectonic reconstructions[41], the Nicaraguan Rise was fully accessible for northward flow only after 9 Ma, which likely explains the convergence of all seawater $\varepsilon_{Nd}$ records between 9.5 and 8 Ma. Moreover, the essentially identical $\varepsilon_{Nd}$ compositions of a Blake Ridge record[28] support an unrestricted export of Caribbean water into the western North Atlantic during this time interval (Fig. 2).

Between 8 and 6.8 Ma, the Florida Strait record again diverges somewhat from the Caribbean records (Fig. 2), which may also have been a result of a hydrographic barrier. However, during this period, the deep Caribbean seawater $\varepsilon_{Nd}$ composition remained constant[10], and the detrital composition of Site 1000 had already shifted towards that of Orinoco sediments, suggesting that the influence of Pacific intermediate waters was at most minor and any observed changes originated from the Atlantic. On the basis of nannofossil assemblages, it was proposed that the north and south Caribbean basins were separated between 8.35 and 3.65 Ma[39], but the seawater $\varepsilon_{Nd}$ records do not support this interpretation given that Sites 998 and 1000 were similar to the south Caribbean Site 999 record for the majority of this time interval[10,34].

The vast majority of modelling simulations find that an enhanced supply of low salinity Pacific waters to the North Atlantic reduces the production of Northern Component Water (NCW, pre-cursor of North Atlantic Deep Water (NADW)) and hence the strength of the AMOC (ref. [5] and references therein). However, Newkirk and Martin[10] found apparently contradicting evidence that the interval of high Pacific water throughflow into the deep Caribbean between ca. 12.4 and 9.5 Ma corresponded to an increase in the production of NCW, as estimated from $\delta^{13}C$ gradients between the Atlantic and the Pacific[31]. Lear et al.[32] also found evidence for a proto-NADW between 12.5 and 10.5 Ma on the basis of cooler benthic Mg/Ca temperatures and higher $\delta^{13}C$ signatures. Our new record from Site 1006 now shows that this interval of high NCW production[30–32] (Fig. 4) occurred at a time when there was little or no export of intermediate waters from the

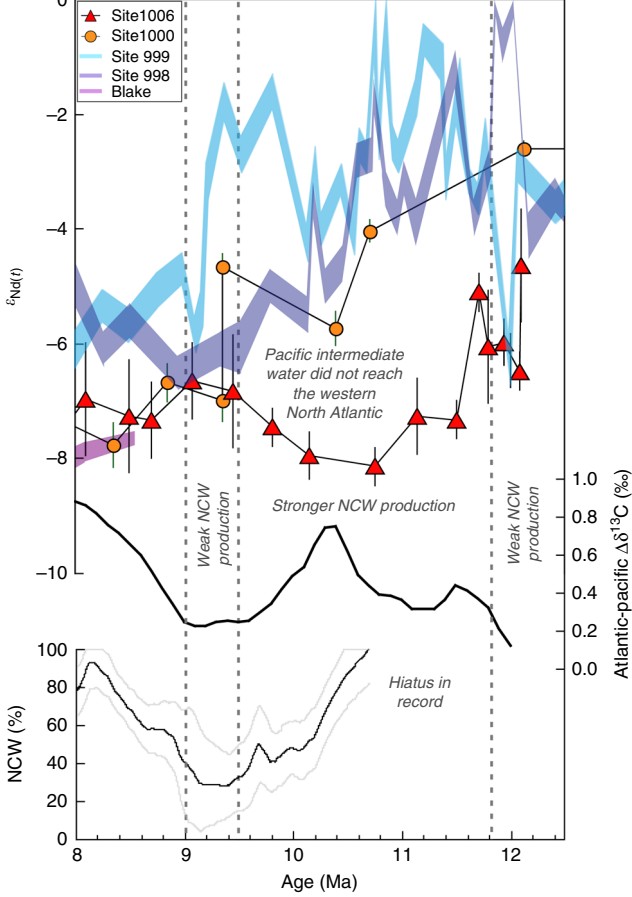

**Fig. 4** Comparison of seawater Nd isotope records with estimates of Northern Component Water production. Seawater $\varepsilon_{Nd(t)}$ signatures of Caribbean and Western Atlantic sites (this study[10, 34]) together with two different estimates of Northern Component Water (NCW) production based on $\delta^{13}C$ gradients between the Atlantic and the Pacific (upper black curve, ref. [32]; lower three curves showing NCW production (thin black curve) and ±1 sigma uncertainty (thin grey lines), with a hiatus and unreliable estimates before 10.7 Ma omitted, ref. [30]) for the period between 12.5 and 8 Ma. Vertical dashed lines show episodes of weak NCW production when the $\varepsilon_{Nd(t)}$ records indicate strong Pacific-western North Atlantic throughflow

Pacific to the western North Atlantic and thus, importantly, the low-salinity Pacific waters present in the Caribbean at that time[10] could not be transferred to the high northern latitudes of the Atlantic (Fig. 5). Following the same reasoning, lower NCW production between 9.5 and 9 Ma may be attributed to the re-establishment of the Pacific-North Atlantic connection, either directly via the Florida Strait or indirectly via the Atlantic gyre (Figs. 4 and 5). These findings support model predictions that changes in the supply of Pacific waters to the North Atlantic affected the AMOC and highlights the importance of intermediate water mass exchange.

According to the results of an $\varepsilon_{Nd}$-enabled ocean–atmosphere general circulation model, the Atlantic became the dominant source of intermediate-depth Caribbean waters once the CAS had shoaled to between 500 and 200 m[6]. Our new records constrain the timing of this significant change away from a Pacific supply to ~9 Ma, therefore providing support to studies that found a major step in CAS closure during the Late Miocene[12–14]. However, the new data do not support a complete closure of the CAS by

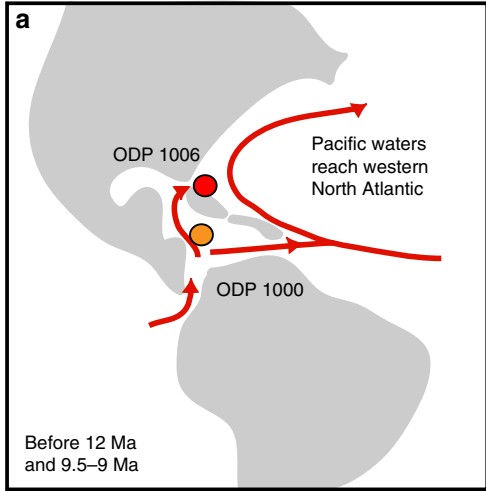

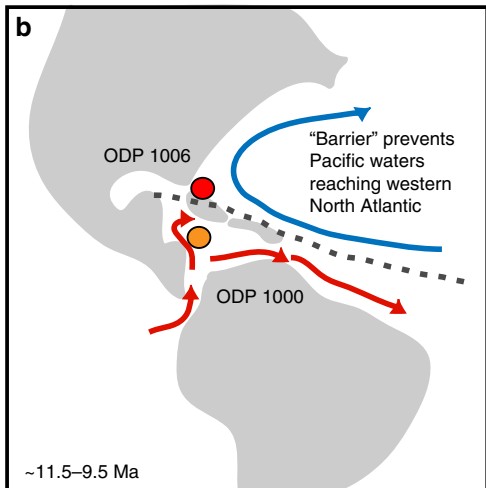

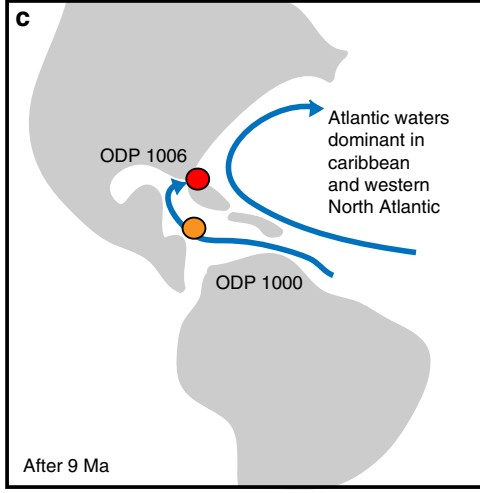

**Fig. 5** Schematic representation of different circulation patterns as the Central American Seaway shoaled. **a** Intermediate-depth water from the Pacific reaches the western North Atlantic, either directly via the Florida Strait, or indirectly via entrainment in the Atlantic gyre. Site 1000 (orange circle) and Site 1006 (red circle) both have radiogenic $\varepsilon_{Nd}$ signatures. **b** A barrier (dashed grey line), either physical or hydrographical, prevents Pacific intermediate waters from reaching the western North Atlantic. Site 1000 has a more radiogenic signal than Site 1006. **c** The Central American Seaway has shoaled sufficiently so that the Atlantic is the dominant source of intermediate waters to the Caribbean. The detrital signal at Site 1000 changes towards that of the Orinoco after 9 Ma

the Florida Strait shows an $\varepsilon_{Nd}$ value of −9 (ref. [42]) as a result of admixture of waters from the western North Atlantic subtropical gyre[46,47]. Similar Nd isotope compositions in the Florida Strait[33] and the Atlantic gyre[47] preclude any calculation of the exact contributions from each source at Site 1006 but low flow speeds in the deep Santaren Channel[45] would allow for mixing between the two. On the Pacific side of the Isthmus of Panama, the $\varepsilon_{Nd}$ signatures of intermediate-depth eastern equatorial Pacific water masses are much more radiogenic (−4 to −1.6)[48].

**Age models**. The age models for Site 1006 and for the Miocene part of Site 1000 are based on published biostratigraphy[49,50].

**Sample preparation**. Sediment samples from ODP Sites 1000A and 1006A were obtained from the IODP Core Repository in Bremen. A further seven core-top samples from the GBB area were obtained from the Seafloor Samples Laboratory of the Woods Hole Oceanographic Institution (Supplementary Fig. 2). All samples were freeze-dried and weighed and an aliquot of 2.5 g of each sample was separated and ground using an agate pestle and mortar. The remaining sample aliquot was wet sieved, oven dried and separated into size fractions ready for microscope work.

Approximately 35 mg of mixed planktonic foraminifera were picked (mostly *Globigerinoides ruber* and *Neogloboquadrina dutertrei*) from the >350-µm size fraction for the extraction of the bottom water Nd isotope composition from the ferromanganese coatings of the uncleaned foraminifera[51]. The shells were crushed between two glass plates to open the chambers, and were then ultrasonicated with alternate rinses of distilled (MQ) water and ethanol to remove any detrital particles. A final visual check for removal of particles was performed under the microscope before dissolution of the foraminifera samples in 0.5 M $HNO_3$.

To extract the detrital fraction, the 2.5 g powdered aliquots were leached following the procedures fully described in refs. [52,53]. Samples were rinsed with distilled (MQ) water, followed by a decarbonation procedure using an acetic acid/ sodium acetate buffer (with a pH around 4). A hydroxylamine hydrochloride solution was used as a leaching agent for removing the Fe–Mn oxyhydroxide fraction, and was buffered with acetic acid/sodium hydroxide to reach the pH value of 4. This initial leach solution was removed and an additional 12 h of leaching with newly added leaching solution was carried out to completely remove any authigenic coatings. After a triple rinse with MQ water and oven drying, an aliquot of 0.05 g of the remaining detrital fraction underwent total dissolution via hotplate digestion using a mixture of $HNO_3$, HF and HCl. Hydrogen peroxide and $HClO_4$ were included in the procedure for the removal of organic material.

Further separation of Nd and Sr and purification of the sample solutions was carried out ion chromatographically. All samples were put through cation exchange columns (AG50W-X12 resin, mesh 200–400 µm)[54] to collect rare earth elements (REEs) and, in the case of the detrital samples, the Sr cut also was collected. Thereafter, Nd was separated from the other REEs (Ln Spec resin columns, mesh 50–100 µm)[55] and Sr was separated from Rb and other interfering elements (Sr spec resin column, mesh 50–100 µm)[56].

**Nd and Sr isotope analysis**. Isotope ratios were measured on a Nu Instruments Multiple Collector Inductively Coupled Plasma Mass Spectrometer (MC-ICPMS) at GEOMAR. The $^{143}Nd/^{144}Nd$ results were mass bias corrected to a $^{146}Nd/^{144}Nd$ ratio of 0.7219 and normalized to the accepted JNdi-1 standard $^{143}Nd/^{144}Nd$ value of 0.512115 (ref. [57]). For downcore samples, $\varepsilon_{Nd(t)}$ values were calculated by correcting the measured $\varepsilon_{Nd(0)}$ for ingrowth of $^{143}Nd$ using an Sm/Nd ratio of 0.139 for uncleaned foraminiferal calcite[58] and 0.109 for detrital material, based on the average upper crustal ratio[59]. The external reproducibility ($2\sigma$) was between 0.21 and 0.40 $\varepsilon_{Nd}$ units for the detrital samples. The external reproducibility ($2\sigma$) for the measurements of the foraminifera samples was higher (0.98 $\varepsilon_{Nd}$ units) due to smaller sample sizes that required a time resolved measurement method.

After correction for Kr and Rb interferences, Sr results were mass bias corrected to an $^{86}Sr/^{88}Sr$ ratio of 0.1194 and normalized to the widely accepted $^{87}Sr/^{86}Sr$ value of 0.710245 for the NIST SRM987 standard. The $^{87}Sr/^{86}Sr$ results were corrected for ingrowth of $^{87}Sr$ using an $^{87}Rb/^{87}Sr$ ratio of bulk earth of 0.09 (ref. [60]). The external reproducibility ($2\sigma$) was between 0.000019 and 0.000022.

10 Ma[14] and instead allow for a shallow seaway or seaways until at least 9 Ma and possibly much later.

## Methods

**Regional setting**. ODP Site 1000 is today bathed by diluted Antarctic Intermediate Water (AAIW)[42] that can be traced by its salinity minimum as far as 20°N in the North Atlantic[43]. ODP Site 1006 is located in the Santaren Channel and receives a mixture of waters from the Caribbean and recirculated gyre waters[44,45] (Fig. 1). AAIW today largely preserves its Atlantic $\varepsilon_{Nd}$ signature of −10.6 to −11 during transit across the Caribbean[42]. Consequently, intermediate-depth waters close to Site 1000 in the Central Caribbean have an $\varepsilon_{Nd}$ value of ~−10, whereas Site 1006 in

Total procedural blanks for Nd chemistry were ≤360 pg and for Sr chemistry were ≤4600 pg. In both cases, these blanks were below 1% of the total sample size and therefore considered insignificant.

## Data availability

Data generated during this study are available in the PANGAEA database (https://doi.org/10.1594/PANGAEA.904251).

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

## Acknowledgements

This work was funded through the M.Sc. Program for Polar and Marine Science POMOR/ Bundesministerium für Bildung und Forschung (BMBF) (V.K.) and German Science Foundation (DFG) project OS 499/2-1 (A.H.O.). Sample material used in the project was provided by the Ocean Drilling Program (ODP) and the Seafloor Samples Laboratory of the Woods Hole Oceanographic Institution. ODP is sponsored by the U.S. National Science Foundation (NSF) and participating countries under management of Joint Oceanographic Institutions (JOI), Inc. We thank R. Samworth and N. Poore for providing additional data for Northern Component Water calculations.

## Author contributions

A.H.O. and M.F. developed the project. V.K., A.H.O. and T.S. performed the isotope analyses. All authors contributed to the writing of the manuscript.

## Additional information

**Competing interests:** The authors declare no competing interests.

