## [Peer Review File · Nature Communications]

Reviewers' comments:

Reviewer #1 (Remarks to the Author):

This is a review of the manuscript „Miocene restriction of the Florida Strait throughflow strengthened Atlantic overturning circulation“ by Kirillova et al. The manuscript uses Neodymium isotopes to reconstruct the history of intermediate water flow from the Pacific through the Panamanian Gateway into the North Atlantic via the Florida Straits using several site locations. The records cover the time period since the middle Miocene showing a consistent pattern of changes through time. Interestingly Pacific water did not seem to have reached the Florida Straits between 9 and 11.5 Ma suggesting that a hydrological barrier existed between the Caribbean and the Florida Straits.

The results of this study are a valuable addition in the ongoing debate of the closure of the Panamanian Gateway and its effects on AMOC. The large differences in ϵ_{Nd} characteristics between the Pacific and the Atlantic provide very clear records of changes through time and are therefore very convincing. Recently, the closing discussion was illustrated by several papers and comments in *Science Advances* in 2017 (O'Dea et al.; Molnar; Jaramillo et al.) following studies that claim that the Panamanian Gateway may have been closed during the middle Miocene already (Montes et al.). I think that one of the main reasons for this ongoing discussion is the matter of definition of closing, if closing a deep or shallow water connection is meant, or which areas, i.e. especially in the Miocene additional Pacific-Atlantic connections were present, are being discussed. Montes et al. only discuss the Panama area and appear to mean with the closing the cessation of deep water connections. In that case, these studies would not oppose the paleoceanographic evidence of the final closure, i.e. surface connections, during the Pliocene. The current study is providing an important piece to this discussion by reconstructing the intermediate flow that is indeed fitting in between the cessation of deep water in the middle Miocene and surface water in the Pliocene. Considering that the manuscript is submitted to *Nature Comm.* I think it would be worth it to include some discussion on this ongoing topic; it would increase the visibility/impact of the manuscript.

In general, I find the manuscript very easy and clear to read, as pointed out before it is providing an important piece of the puzzle on a topic that is being discussed for decades already. Accordingly, I do not have any major comments to this and would recommend to publish in *Nature Comm.* with minor revisions. Some minor comments follow below.

As mentioned earlier I would like to see some discussion relating to do ongoing discussion that some studies now claim that the Panamanian Gateway may have been closed in the middle Miocene already. It is discussed in the manuscript already how the Nicaraguan Rise developed into the late Miocene. But have the authors considered how a possible impact of a gateway north of the Rise may have affected their records? The Rise is part of the Chortis Block and its northern boundary may have been an existing gateway during the Miocene (Coates and Obando, 1996; Stehli and Webb, 1985).

The hydrological barrier preventing a northward flow from the Caribbean to the Florida Straits makes sense to have existed. Is it possible however that the Pacific water may have been included into the Atlantic gyre and then still reached Blake plateau (and possibly also the Bahamas)? Are there any modelling studies that hint towards this? Because, if Pacific water continued to flow eastward into the tropical Atlantic, which water mass was supplying the AMOC/NADW formation in the north Atlantic?

In the supplements it is mentioned that the contribution of more radiogenic waters from the South Atlantic would have need to be "huge" to explain the signals reconstructed. Can you give an estimate for this instead of just "huge"?

Reviewer #2 (Remarks to the Author):

This manuscript presents high quality Nd isotope data from two sites in the Caribbean. The authors provide evidence supporting conclusions that were drawn from floral assemblages in a 2000 paper (ref. 40), that in the middle Miocene there was a hydrographic barrier preventing exchange of seawater between the southern and northern Caribbean. As most of the outflow from the Caribbean enters the Gulf Stream once it passes through the Florida straits, small changes in salinity have a large impact on the production of deep water in the North Atlantic, the descending limb of the Atlantic Meridional Overturning Circulation (AMOC). In the middle Miocene, the Central American Seaway (CAS) was open, allowing relatively low salinity Pacific waters to become entrained in Caribbean waters that ultimately entered the North Atlantic via the Florida straits. The Nd isotope data presented here provide a more robust way to evaluate the temporal linkages between the export of Caribbean water (and the low salinity water the Caribbean received from the Pacific). The divergence of ϵNd values between the northern and southern sites in this study in the middle Miocene coincides with an increase in northern component water (NCW) export.

This is an interesting contribution and will certainly add to the heady bouquet of interpretations surrounding the interplay of CAS tectonics, Caribbean outflow, and AMOC dynamics. I have made some suggestions below to improve the manuscript.

1. Main text

Line 135-137: Could the authors be convinced to provide a sketch of what this hydrographic barrier looks like? It is difficult to imagine this mechanism. Moreover, the authors can utilize ref. 40 more effectively to support the interpretation in this manuscript. The floral response to the circum-tropical current is exactly what the authors are describing. The way the text reads, it seems like ref. 40 is simply being invoked to supply a mechanism to separate the northern and southern Caribbean.

Line 169-173: Is it possible that circulation changes resulted in the Caribbean outflow to bypass the Florida straits and flow into the North Atlantic further to the east?

2. References

Citation for ref. 42 needs editing.

3. Figures

In addition to labeling the site numbers, I suggest labeling site 1000 as 'southern' and site 1006 as 'northern' to aid in the interpretation of this diagram.

4. Supplemental material

Lines 31-34: was the detrital digestion done in steel jacketed bombs or was it a hotplate digestion?

Line 66: interpretation

Line 70: change to "compared to the Caribbean"

Line 95-98: Isn't it also a possibility that divergence between 1006 and 1000 indicates less Caribbean sourced waters exiting through the Florida straits without any change in Pacific CAS throughflow? Is the Florida strait pathway the only way to get Caribbean entrained waters into

Reviewer #3 (Remarks to the Author):

Kirilova et al.'s manuscript provides new ϵNd records at two sites close to and in the Caribbean Sea, expanding in time previous Miocene records. From this new data, the literature, and a direct

comparison with previously published Northern Component Water (%NCW) reconstructions, authors discuss the synchronicity between Caribbean flow in the Florida Strait, Central American Seaway constriction, and Atlantic Meridional Overturning Circulation dynamics. The manuscript is (very) concise, and mostly well-written. The figures are of high quality, making it easy for the reviewer to navigate through the manuscript and evaluate the discussion.

My first point is that it is very important for the paleoceanography community to get this data published, as expanding these records in time is crucial to better understand the ocean dynamics in the region during the Miocene, a period marked by several important tectonics and climate changes. In that regards, the results are well described and again, easy to visualize. Not being a data scientist, I am not able to evaluate the methods section in the SI, the robustness of the analysis (and the absence of error bars on the curves) but do acknowledge the expanded discussion about potential mechanisms in this same SI.

Unfortunately, the quality of the discussion within the MS does not align with the quality of the result section.

1/ The authors make one major interpretation and claim from their results; that water exported from the Florida Strait controlled AMOC before 9 Ma, more specifically that an interruption of intermediate water mass export through this strait resulted in a stronger AMOC between 11.5 and 9.5 Ma. This interpretation mostly relies on the comparison of the ϵ_{Nd} record at site 1006 and %NCW reconstructions from Poore et al. (2006, G3), that are reported on figure 4. This is a major concern for me: I cannot find the same data in the cited Poore et al. paper. In their paper, Poore et al., after insisting on the caveats of the methods to be interpreted in terms of AMOC strength, show a hiatus between 10.7 and 11.5 Ma (their figure 10), even highlighting that %NCW older than 10.7 Ma cannot be considered as robust (their figure 9). That means that one cannot state anything robust about the tendencies in %NCW (and AMOC !) for this particular period. Still, Kirilova et al. seem ignore this hiatus by simply linearly interpolating Poore et al. values between 11.5 and 10.7 Ma on their figure 4, which does not make sense at all to me if not justified. Lastly, the entire %NCW envelope seems awkwardly expanded towards more negative value when compared to Poore et al. figure 10 (see for example the min values between 11.5 and 12 Ma : around 40% in Poore et al., around 15% in the ms), again with no justification. Also, Poore et al. data shows that %NCW is decreasing between 10.7 and 9.5 Ma, contradicting Kirilova et al. statements. Thus the entire discussion about the link between ϵ_{Nd} record and AMOC collapses.

2/ Authors evoke several times the balance between the two water masses that site 1006 actually records. I think this question is crucial. Rapidly looking at NOAA current velocity data and the location of site 1006 in the Santareen channel, it appears, as authors acknowledge, that 1006 integrates a subtle mix of Florida-Strait-originating water masses and western Atlantic gyre water masses. Also, (low) current velocity (and hence long residence time) here together with margin ϵ_{Nd} -values will likely influence the ultimate ϵ_{Nd} value. Authors should discuss more this balance. What do we know about the local paleogeography, what makes them think it is a 50/50 balance, or not ? The signal and its interpretation is highly dependent on that. If site 1006 capture mostly Atlantic water masses, the strength of the water throughflow through the Florida Strait, which will very likely be lower than present with an open CAS (see modelling papers), does not matter much, and thus the ϵ_{Nd} can be interpreted just in terms of Atlantic gyre dynamics and Pacific waters inputs, not through the FS, but through the southern Antiles and recapture by the Atlantic gyre. Discarding %NCW data and considering the location of site 1006, a more parcimonious interpretation of these new records seems to be that the ongoing shoaling of CAS progressively lowered the input of intermediate Pacific waters in the Atlantic, as testify by the overall lowering of ϵ_{Nd} signature from 12 Ma to present. Short timescales variations should be taken carefully : I agree they could trace changes in AMOC activity (although the comparison with Moor et al. paper needs to be solved), but I remind that AMOC at that time was under the influence of several other drivers that CAS (see for ex. Brierley & Fedorov, 2016), including various uplifts that ultimately alter Evaporation/precipitation balances and ocean dynamics, the East-Tethys closure that controls input of freshwater in the Atlantic, and major changes in the Amazon routing system that shifts river inputs from the Caribbean to the tropical Atlantic (Hoorn et al., 2010).

Convergence of 1006 & 1000 values at 9 Ma seems to clearly mark the shutdown of Pacific massive input in the Caribbean, which is consistent with the timing suggested for a shallow seaway in Sepulchre et al. (2014).

3/ I think some of the modelling literature regarding the CAS impact on water masses fates in the Atlantic has been overlooked. Authors cite Pfister et al. (2014), which is a paper using low resolution model of intermediate complexity, and can hardly provide information about water masses portioning in the area. Sepulchre et al. (2014) made a review of the papers dealing with an open CAS, and showed the various responses of the AMOC, Florida Strait flow, and more importantly $\epsilon\text{-Nd}$ values to CAS depth and initial conditions. Nof & Van Gorder (2003) showed that the dynamics in the region as a response to CAS closure also is very dependent on initial conditions.

Less crucial remarks :

- The CAS closure debate is not very well presented. Authors should evoke the original papers by Montes et al. (with the Zircon indicator) and Bacon et al. (2015, PNAS) paper using the original phylogenetic approach to comfort the idea of an early closure of the CAS. See also Hoorn & Flantua (Science, 2015).
- Authors evoke the potential influence of river discharge on the signal. That's a very interesting point that would need more discussion regarding the recent advances in the fate of the Amazon river basins during the Miocene as a response to Andes uplift.

Overall recommendation: Reject with invitation to resubmit.

Response to Reviewers for:

Miocene restriction of the Pacific-North Atlantic throughflow strengthened Atlantic overturning circulation

Valeriia Kirillova^a, Anne H. Osborne^{a,*}, Tjördis Störli^a, and Martin Frank^a

^aGEOMAR Helmholtz Centre for Ocean Research Kiel, Germany.

*Corresponding author: aosborne@geomar.de

Reviewers' comments:

Reviewer #1 (Remarks to the Author):

This is a review of the manuscript „Miocene restriction of the Florida Strait throughflow strengthened Atlantic overturning circulation“ by Kirillova et al. The manuscript uses Neodymium isotopes to reconstruct the history of intermediate water flow from the Pacific through the Panamanian Gateway into the North Atlantic via the Florida Straits using several site locations. The records cover the time period since the middle Miocene showing a consistent pattern of changes through time. Interestingly Pacific water did not seem to have reached the Florida Straits between 9 and 11.5 Ma suggesting that a hydrological barrier existed between the Caribbean and the Florida Straits.

The results of this study are a valuable addition in the ongoing debate of the closure of the Panamanian Gateway and its effects on AMOC. The large differences in ϵ_{Nd} characteristics between the Pacific and the Atlantic provide very clear records of changes through time and are therefore very convincing. Recently, the closing discussion was illustrated by several papers and comments in Science Advances in 2017 (O’Dea et al.; Molnar; Jaramillo et al.) following studies that claim that the Panamanian Gateway may have been closed during the middle Miocene already (Montes et al.). I think that one of the main reasons for this ongoing discussion is the matter of definition of closing, if closing a deep or shallow water connection is meant, or which areas, i.e. especially in the Miocene additional Pacific-Atlantic connections were present, are being discussed. Montes et al. only discuss the Panama area and appear to mean with the closing the cessation of deep water connections. In that case, these studies would not oppose the paleoceanographic evidence of the final closure, i.e. surface connections, during the Pliocene. The current study is providing an important piece to this discussion by reconstructing the intermediate flow that is indeed fitting in between the cessation of deep water in the middle Miocene and surface water in the Pliocene. Considering that the manuscript is submitted to Nature Comm. I think it would be worth it to include some discussion on this ongoing topic; it would increase the visibility/impact of the manuscript.

The debate on the evolution of the Gateway closure at different water depths is now a more prominent part of the manuscript. The ongoing discussion is presented in more detail in the introduction, and the importance of our new data for the debate is emphasized at the end of the manuscript.

In general, I find the manuscript very easy and clear to read, as pointed out before it is providing an important piece of the puzzle on a topic that is being discussed for decades

already. Accordingly, I do not have any major comments to this and would recommend to publish in Nature Comm. with minor revisions. Some minor comments follow below.

We appreciate the positive feedback of the reviewer concerning the importance of our results.

As mentioned earlier I would like to see some discussion relating to do ongoing discussion that some studies now claim that the Panamanian Gateway may have been closed in the middle Miocene already.

See previous reply

It is discussed in the manuscript already how the Nicaraguan Rise developed into the late Miocene. But have the authors considered how a possible impact of a gateway north of the Rise may have affected their records? The Rise is part of the Chortis Block and its northern boundary may have been an existing gateway during the Miocene (Coates and Obando, 1996; Stehli and Webb, 1985).

We find no discussion of the northern boundary of the Chortis Block in the Coates and Obando (1996) publication. There are two chapters in the book edited by Stehli and Webb (1985) that include information about the position of the Chortis block. Smith (1985) shows an active fault and possible seaway north of the Chortis Block at approximately 30 Ma, based on the work of Wadge and Burke (1983). However, the possibly existing seaway is not present in the early Oligocene reconstruction based on the work of Malfait and Dinkelman (1972) nor is it present in the Miocene reconstruction based on the work of Anderson and Schmidt (1983). In the other chapter of Stehli and Webb (1985) that references the Chortis Block, Donnelly (1985) presents a figure which shows a much earlier collision of the Chortis Block and the Yucatan-North American Block along the Motagua Suture Zone at the end Cretaceous at approximately 65 Ma.

Given that we did not find any evidence for a Miocene gateway north of the Nicaraguan Rise in the literature, we decided against discussing this in the text.

The hydrological barrier preventing a northward flow from the Caribbean to the Florida Straits makes sense to have existed. Is it possible however that the Pacific water may have been included into the Atlantic gyre and then still reached Blake plateau (and possibly also the Bahamas)? Are there any modelling studies that hint towards this? Because, if Pacific water continued to flow eastward into the tropical Atlantic, which water mass was supplying the AMOC/NADW formation in the north Atlantic?

We now discuss both a direct connection between the Pacific and the Bahamas via the Florida Strait and the possible entrainment of Pacific waters into the Atlantic gyre via an alternative route. The Sepulchre et al. (2014) modelling results suggest that entrainment into the gyre may have occurred, which is now included in the discussion. A new figure (5) illustrates that both routes may have existed. This does, however, not question our overall observation that changes in the supply of Pacific waters to the western North Atlantic, either via the Florida Strait or via the gyre, co-varied with AMOC strength, in line with model predictions (eg. Zhang et al. (2012) and references therein).

In the supplements it is mentioned that the contribution of more radiogenic waters from the

South Atlantic would have need to be “huge” to explain the signals reconstructed. Can you give an estimate for this instead of just “huge”?

This section has been reworded to emphasise that evidence for Southern Ocean end-member ϵ_{Nd} compositions between -10 and -7 over the past 14 Myr excludes advection of more radiogenic waters from the South Atlantic as an explanation for the signals detected without invoking a Pacific contribution via the Central America Seaway.

Reviewer #2 (Remarks to the Author):

This manuscript presents high quality Nd isotope data from two sites in the Caribbean. The authors provide evidence supporting conclusions that were drawn from floral assemblages in a 2000 paper (ref. 40), that in the middle Miocene there was a hydrographic barrier preventing exchange of seawater between the southern and northern Caribbean. As most of the outflow from the Caribbean enters the Gulf Stream once it passes through the Florida straits, small changes in salinity have a large impact on the production of deep water in the North Atlantic, the descending limb of the Atlantic Meridional Overturning Circulation (AMOC). In the middle Miocene, the Central American Seaway (CAS) was open, allowing relatively low salinity Pacific waters to become entrained in Caribbean waters that ultimately entered the North Atlantic via the Florida straits. The Nd isotope data presented here provide a more robust way to evaluate the temporal linkages between the export of Caribbean water (and the low salinity water the Caribbean received from the Pacific. The divergence of ϵ_{Nd} values between the northern and southern sites in this study in the middle Miocene coincides with an increase in northern component water (NCW) export.

This is an interesting contribution and will certainly add to the heady bouquet of interpretations surrounding the interplay of CAS tectonics, Caribbean outflow, and AMOC dynamics. I have made some suggestions below to improve the manuscript.

1. Main text

Line 135-137: Could the authors be convinced to provide a sketch of what this hydrographic barrier looks like? It is difficult to imagine this mechanism.

A new figure has been added (Figure 5) to illustrate the major features of Caribbean and Atlantic circulation changes indicated by the new records. Included in part (b) is an estimation of where the hydrographic (or possibly physical) barrier may have been located. We cannot be more precise based on current evidence.

Moreover, the authors can utilize ref. 40 more effectively to support the interpretation in this manuscript. The floral response to the circum-tropical current is exactly what the authors are describing. The way the text reads, it seems like ref. 40 is simply being invoked to supply a mechanism to separate the northern and southern Caribbean.

The findings of Kameo and Sato (2000) are now explicitly referred to in the text.

Line 169-173: Is it possible that circulation changes resulted in the Caribbean outflow to bypass the Florida straits and flow into the North Atlantic further to the east?

See reply to Reviewer #1 and Figure 5, which now exactly illustrate this pattern.

2. References

Citation for ref. 42 needs editing.

This is a glitch in the EndNote software and will be edited in the final version

3. Figures

In addition to labeling the site numbers, I suggest labeling site 1000 as ‘southern’ and site 1006 as ‘northern’ to aid in the interpretation of this diagram.

We disagree, as the Sites are not referred to as such in the text.

4. Supplemental material

Lines 31-34: was the detrital digestion done in steel jacketed bombs or was it a hotplate digestion?

This was a hotplate digestion and this information is now included in the supplementary information.

Line 66: interpretation

This has been changed to “interpretation”.

Line 70: change to “compared to the Caribbean”

This has been changed to “compared to the Caribbean”

Line 95-98: Isn’t it also a possibility that divergence between 1006 and 1000 indicates less Caribbean sourced waters exiting through the Florida straits without any change in Pacific CAS throughflow?

This is exactly what we are suggesting!

Is the Florida strait pathway the only way to get Caribbean entrained waters into

(Reviewer’s sentence incomplete?)

No, the pathway via the gyre is also a possibility for Pacific waters to reach Site 1006 and this is now included in the discussion but this does not change the overall interpretation of the records.

Reviewer #3 (Remarks to the Author):

Kirilova et al.’s manuscript provides new eps-Nd records at two sites close to and in the Caribbean Sea, expanding in time previous Miocene records. From this new data, the literature, and a direct comparison with previously published Northern Component Water (%NCW) reconstructions, authors discuss the synchronicity between Caribbean flow in the Florida Strait, Central American Seaway constriction, and Atlantic Meridional Overturning Circulation dynamics. The manuscript is (very) concise, and mostly well-written. The figures

are of high quality, making it easy for the reviewer to navigate through the manuscript and evaluate the discussion.

My first point is that it is very important for the paleoceanography community to get this data published, as expanding these records in time is crucial to better understand the ocean dynamics in the region during the Miocene, a period marked by several important tectonics and climate changes. In that regards, the results are well described and again, easy to visualize. Not being a data scientist, I am not able to evaluate the methods section in the SI, the robustness of the analysis (and the absence of error bars on the curves) but do acknowledge the expanded discussion about potential mechanisms in this same SI.

We have omitted the error bars for each data point on Figure 2 to improve clarity but have included the error bars in figure S3. This is now mentioned in the caption of Figure 2.

Unfortunately, the quality of the discussion within the MS does not align with the quality of the result section.

1/ The authors make one major interpretation and claim from their results; that water exported from the Florida Strait controlled AMOC before 9 Ma, more specifically that an interruption of intermediate water mass export through this strait resulted in a stronger AMOC between 11.5 and 9.5 Ma. This interpretation mostly relies on the comparison of the eps-Nd record at site 1006 and %NCW reconstructions from Poore et al. (2006, G3), that are reported on figure 4. This is a major concern for me: I cannot find the same data in the cited Poore et al. paper. In their paper, Poore et al., after insisting on the caveats of the methods to be interpreted in terms of AMOC strength, show a hiatus between 10.7 and 11.5 Ma (their figure 10), even highlighting that %NCW older than 10.7 Ma cannot be considered as robust (their figure 9). That means that one cannot state anything robust about the tendencies in %NCW (and AMOC !) for this particular period. Still, Kirilova et al. seem ignore this hiatus by simply linearly interpolating Poore et al. values between 11.5 and 10.7 Ma on their figure 4, which does not make sense at all to me if not justified. Lastly, the entire %NCW envelope seems awkwardly expanded towards more negative value when compared to Poore et al. figure 10 (see for example the min values between 11.5 and 12 Ma : around 40% in Poore et al., around 15% in the ms), again with no justification.

We agree that the initial version of Figure 4 did not correctly reflect the presence of the hiatus and the uncertainties outlined in the Poore et al. paper for their record prior to 10.7 Ma. Figure 4 has been corrected and now only shows the Poore et al. (2006) data from 10.7 Ma onwards. The plot of the Poore et al. data is now also correctly represented by the corresponding y-axis, which had accidentally been displaced in Figure 4 of the initially submitted manuscript. We also added an additional and independent proxy record of AMOC strength to Figure 4, which covers the entire period of time from 12 to 8 Ma (Lear et al., 2003) and which supports our interpretations.

Also, Poore et al. data shows that %NCW is decreasing between 10.7 and 9.5 Ma, contradicting Kirilova et al. statements. Thus the entire discussion about the link between eps-Nd record and AMOC collapses.

We disagree. Our argument is that the AMOC was stronger between 11.5 and 9.5 Ma than it was before 12 Ma and between 9.5 and 9 Ma. The Poore et al. (2006) record supported by the one of Lear et al. (2003) (both based on $\Delta\delta^{13}C$) both clearly indicate that the AMOC was stronger prior to 9.5 Ma than between 9.5 and 9 Ma. The newly added Lear et al. (2003)

record extends back to ~ 12 Ma, and confirms this evolution of AMOC strength in that it only shows a small Atlantic-Pacific $\Delta\delta^{13}C$ between ~9.5 and 9 Ma as well as prior to ~ 11.8 Ma reflecting a weaker AMOC.

2/ Authors evoke several times the balance between the two water masses that site 1006 actually records. I think this question is crucial. Rapidly looking at NOAA current velocity data and the location of site 1006 in the Santareen channel, it appears, as authors acknowledge, that 1006 integrates a subtle mix of Florida-Strait-originating water masses and western Atlantic gyre water masses. Also, (low) current velocity (and hence long residence time) here together with margin eps-values will likely influence the ultimate eps-Nd value. Authors should discuss more this balance. What do we know about the local paleogeography, what makes them think it is a 50/50 balance, or not? The signal and its interpretation is highly dependent on that. If site 1006 capture mostly Atlantic water masses, the strength of the water throughflow through the Florida Strait, which will very likely be lower than present with an open CAS (see modelling papers), does not matter much, and thus the eps-Nd can be interpreted just in terms of Atlantic gyre dynamics and Pacific waters inputs, not through the FS, but through the southern Antiles and recapture by the Atlantic gyre. Discarding %NCW data and considering the location of site 1006, a more parcimonious interpretation of these new records seems to be that the ongoing shoaling of CAS progressively lowered the input of intermediate Pacific waters in the Atlantic, as testify by the overall lowering of eps-Nd signature from 12 Ma to present.

We have added information to the section “Regional Setting” to emphasise what we can and cannot say about the exact mixture of waters reaching Site 1006. Most important for the discussion is whether or not Pacific waters reached Site 1006 and it does not change our interpretation if those waters were delivered ‘directly’ via the Florida Strait, or ‘indirectly’ via entrainment in the Atlantic gyre.

The sediment of Site 1006 generally consists of > 80 % pelagic and bank-derived carbonate but the carbonate percentage in some clay-rich layers was as low as 50 % (Eberli et al. 1997). The carbonate is expected to have an unradiogenic ϵ_{Nd} signature as these formed in ambient seawater. In contrast, the clays are thought to originate from Cuba and Hispanola (Eberli et al. 1997) and are expected to have a more radiogenic Nd isotope composition. However, these clay rich layers (which were avoided during sampling) are concentrated in the upper 130 m of the core (Pleistocene and younger), which show the least radiogenic extracted seawater ϵ_{Nd} signal. Furthermore, our measured core-top ϵ_{Nd} signature agrees well with that of ambient seawater. We therefore are convinced that the authigenic Nd isotope signatures of Site 1006 reliably reflect past seawater compositions.

Short timescales variations should be taken carefully : I agree they could trace changes in AMOC activity (although the comparison with Moor et al. paper needs to be solved), but I remind that AMOC at that time was under the influence of several other drivers that CAS (see for ex. Brierley & Fedorov, 2016), including various uplifts that ultimately alter Evaporation/precipitation balances and ocean dynamics, the East-Tethys closure that controls input of freshwater in the Atlantic, and major changes in the Amazon routing system that shifts river inputs from the Caribbean to the tropical Atlantic (Hoorn et al., 2010).

It is beyond the scope of this (short) manuscript to discuss the various possible factors that forced the AMOC. Our main finding is that there was a reduction in the amount of Pacific water reaching the western North Atlantic during a period of increased AMOC strength, in line with model predictions. Our new records support this simple and direct driver for

circulation change in the Atlantic and do not provide any additional evidence for or against other possible factors. Therefore we do not think that our manuscript would be improved by including a discussion of all these other possible drivers. However, we do take on board that the rerouting of the Amazon drainage system may have impacted the detrital material reaching Site 1000, which is now included in the discussion and in fact underlines that a major change in Caribbean circulation caused by separation from Pacific input took place after 9 Ma.

Convergence of 1006 & 1000 values at 9 Ma seems to clearly mark the shutdown of Pacific massive input in the Caribbean, which is consistent with the timing suggested for a shallow seaway in Sepulchre et al. (2014).

We now directly compare our results with the modelling paper of Sepulchre et al (2014) and use this to constrain the depth of the CAS from 9 Ma onwards.

3/ I think some of the modelling literature regarding the CAS impact on water masses fates in the Atlantic has been overlooked. Authors cite Pfister et al. (2014), which is a paper using low resolution model of intermediate complexity, and can hardly provide information about water masses partitioning in the area. Sepulchre et al. (2014) made a review of the papers dealing with an open CAS, and showed the various responses of the AMOC, Florida Strait flow, and more importantly $\epsilon\text{-Nd}$ values to CAS depth and initial conditions.

This is a justified criticism. We now specifically use the results of the Sepulchre et al (2014) model runs to interpret our new records, which enables us to constrain the timing of CAS closure to less than 500 m water depth from ~ 9 Ma.

Nof & Van Gorder (2003) showed that the dynamics in the region as a response to CAS closure also is very dependent on initial conditions.

The calculations of Nof & Van Gorder (2003) found that there should have been a westward flow through the CAS from the Atlantic to the Pacific. However, records from the Pacific show no change in ϵNd consistent with this scenario (see Figure 3, Newkirk & Martin 2009), therefore we do not include it in the discussion.

Less crucial remarks :

- The CAS closure debate is not very well presented. Authors should evoke the original papers by Montes et al. (with the Zircon indicator) and Bacon et al. (2015, PNAS) paper using the original phylogenetic approach to comfort the idea of an early closure of the CAS. See also Hoorn & Flantua (Science, 2015).

A more extensive discussion of the evolution of CAS closure is now included (see reply to Reviewer #1).

- Authors evoke the potential influence of river discharge on the signal. That's a very interesting point that would need more discussion regarding the recent advances in the fate of the Amazon river basins during the Miocene as a response to Andes uplift.

The impact of changes in uplift of the Andes and Amazon river re-routing is now included in the discussion.

Overall recommendation: Reject with invitation to resubmit.

References included in the reply to reviewers

Anderson, T. H., and V. A. Schmidt (1983), The evolution of Middle America and the Gulf of Mexico-Caribbean Sea during Mesozoic time, *Geological Society of America Bulletin*, 94(8), 941-966.

Coates, A. G., and J. A. Obando (1996), The geological evolution of the Central American isthmus, in *Evolution and environment in tropical America*, edited by J. B. C. Jackson, A. F. Budd and A. G. Coates, pp. 21-56, Chicago University, Chicago.

Donnelly, T. W. (1985), Mesozoic and Cenozoic Plate Evolution of the Caribbean Region, in *The Great American Biotic Interchange*, edited by F. G. Stehli and D. J. Webb, pp. 89-121, Plenum Press, New York and London.

Lear CH, Rosenthal Y, Wright JD. The closing of a seaway: ocean water masses and global climate change. *Earth and Planetary Science Letters* **210**, 425-436 (2003).

Malfait, B. T., and M. G. Dinkelman (1972), Circum-Caribbean tectonic and igneous activity and evolution of the Caribbean Plate, *Geological Society of America Bulletin*, 83(2), 251-+.

Newkirk DR, Martin EE. Circulation through the Central American Seaway during the Miocene carbonate crash. *Geology* 37, 87-90 (2009).

Nof D, Van Gorder S. Did an open Panama Isthmus correspond to an invasion of Pacific water into the Atlantic? *Journal of Physical Oceanography* 33, 1324-1336 (2003).

Poore HR, Samworth R, White NJ, Jones SM, McCave IN. Neogene overflow of Northern Component Water at the Greenland-Scotland Ridge. *Geochemistry Geophysics Geosystems* **7**, (2006).

Sepulchre P, *et al.* Consequences of shoaling of the Central American Seaway determined from modelling Nd isotopes. *Paleoceanography*, (2014).

Smith, D. L. (1985), Caribbean Plate Relative Motions, in *The Great American Biotic Interchange*, edited by F. G. Stehli and D. J. Webb, pp. 17-48, Plenum Press, New York and London.

Stehli, F. G., and D. J. Webb (Eds.) (1985), *The Great American Biotic Interchange*, 532 pp., Plenum Press, New York and London.

Wadge, G., and K. Burke (1983), Neogene Caribbean plate rotation and associated Central American tectonic evolution, *Tectonics*, 2(6), 633-643.

Reviewers' comments:

Reviewer #1 (Remarks to the Author):

This is my second review of the paper by Kirillova et al. It was interesting to read that several of the same points were brought up by more than one reviewer like including the recent literature discussion on an early closure of Panama, and especially, one of the most important comments, on the possibility of Pacific water reaching the West Atlantic by being incorporated into the gyre. The authors have clearly and extensively answered these issues. And as they point out, allowing Pacific water into the gyre and then to the Site 1006 location does not change the conclusions of the manuscript, but it does make story and its possible explanation more complete. So this is a very good addition.

Reviewer 3 had more extensive and critical comments including the use of the previous data by Poore et al. that would not allow the current interpretation of changes in NCW estimates. The authors have adapted and corrected their representation of these data and added a new dataset by Lear et al. to support their story. This is also now further discussed by including the extensive modelling studies by Sepulchre et al. that are supporting the results.

I agree with the authors that this manuscript should not go into all the causes and effects of what changed Miocene AMOC, apart from including the switch in the delta of the Amazon that may have had an impact on the Caribbean waters. One thing I am curious about is what the effect of the switch of the Amazon would really have on the Caribbean. When it switched away from the Caribbean itself sediment supply from the higher Andes increased and in the modern situation the Amazon Plume is turning northwestward into the Caribbean. So unless the throughflow was still strong enough to divert Amazon water into the open Atlantic, I think the situation may not have changed that much.

In the last section it is written that the 9 Ma may correspond with other studies that indicate changes in the Mid-Miocene. But 9 Ma is not Middle Miocene; this is already the Late Miocene. According to the Geological Timescale the boundary is at ~11.5 Ma at the Serravalian-Tortonian boundary.

I think this manuscript is now ready for publication with Nature Communications.

Reviewer #4 (Remarks to the Author):

Overall I find think this manuscript present some interesting new data about the closure of the Central American Seaway. I see no substantial impediment to publication.

I have come in as a replacement for Reviewer #3. Similarly to Reviewer #3, I am a data-specialist and so cannot judge those aspects of the research. Therefore, I've focussed my attention to how the authors have responded to the comments of Reviewer #3.

There is now sufficient information about (2) namely the provenance and interpretation of Site 1006, and (3) modelling the palaeoAMOC.

The authors state that other factors potentially influencing AMOC (as brought in point 2 by Reviewer #3) is outside the scope of this manuscript. I agree. But nonetheless I feel that by not even acknowledging this possibility, excessive focus is placed on the CAS closure. Given you rely on a reconstruction of AMOC which is explicitly connected with a different mechanisms (the Greenland-Scotland Ridge), this seems unwarranted. A further caveat sentence would be beneficial.

My main worry relates to the Poore et al (2006) data shown in Fig. 4. Reviewer #3 stated (as their 1st comment) that it did not look like the data shown by Poore et al in either Fig 9 or Fig 10. Whilst this has been edited since, it still does not appear to a direct replicate of that data. The

black line could be correct, but the error bars descending to 0% and having a peak upto 100% at 9Ma are visibly wrong. I feel that the authors have dealt with the rest of the reviewer's comment (by including the Lear et al data).

I had to look up what Northern Component Water was. The manuscript would benefit from a statement that is the palaeo-name for NADW.

Chris Brierley (UCL Geography)

Reviewers' comments:

Reviewer #1 (Remarks to the Author):

This is my second review of the paper by Kirillova et al. It was interesting to read that several of the same points were brought up by more than one reviewer like including the recent literature discussion on an early closure of Panama, and especially, one of the most important comments, on the possibility of Pacific water reaching the West Atlantic by being incorporated into the gyre. The authors have clearly and extensively answered these issues. And as they point out, allowing Pacific water into the gyre and then to the Site 1006 location does not change the conclusions of the manuscript, but it does make story and its possible explanation more complete. So this is a very good addition.

Reviewer 3 had more extensive and critical comments including the use of the previous data by Poore et al. that would not allow the current interpretation of changes in NCW estimates. The authors have adapted and corrected their representation of these data and added a new dataset by Lear et al. to support their story. This is also now further discussed by including the extensive modelling studies by Sepulchre et al. that are supporting the results.

We appreciate the positive feedback of the reviewer regarding the changes we made in the revised manuscript.

I agree with the authors that this manuscript should not go into all the causes and effects of what changed Miocene AMOC, apart from including the switch in the delta of the Amazon that may have had an impact on the Caribbean waters. One thing I am curious about is what the effect of the switch of the Amazon would really have on the Caribbean. When it switched away from the Caribbean itself sediment supply from the higher Andes increased and in the modern situation the Amazon Plume is turning northwestward into the Caribbean. So unless the throughflow was still strong enough to divert Amazon water into the open Atlantic, I think the situation may not have changed that much.

We agree that the main source of detrital material to Site 1000 after 9 Ma was from the Orinoco and Amazon basins, despite the re-routing of the drainage systems away from the Caribbean and into the tropical Atlantic. We have now included the information on re-routing for completeness, in response to comments from Reviewer #3.

In the last section it is written that the 9 Ma may correspond with other studies that indicate changes in the Mid-Miocene. But 9 Ma is not Middle Miocene; this is already the Late

Miocene. According to the Geological Timescale the boundary is at ~11.5 Ma at the Serravalian-Tortonian boundary.

This has now been changed to Late Miocene. Another instance of Mid-Miocene in the third paragraph has been changed to Middle to Late Miocene.

I think this manuscript is now ready for publication with Nature Communications.

Reviewer #4 (Remarks to the Author):

Overall I find think this manuscript present some interesting new data about the closure of the Central American Seaway. I see no substantial impediment to publication.

I have come in as a replacement for Reviewer #3. Similarly to Reviewer #3, I am a data-specialist and so cannot judge those aspects of the research. Therefore, I've focussed my attention to how the authors have responded to the comments of Reviewer #3.

There is now sufficient information about (2) namely the provenance and interpretation of Site 1006, and (3) modelling the palaeoAMOC.

The authors state that other factors potentially influencing AMOC (as brought in point 2 by Reviewer #3) is outside the scope of this manuscript. I agree. But nonetheless I feel that by not even acknowledging this possibility, excessive focus is placed on the CAS closure. Given you rely on a reconstruction of AMOC which is explicitly connected with a different mechanisms (the Greenland-Scotland Ridge), this seems unwarranted. A further caveat sentence would be beneficial.

Reviewer #3 asked us to consider other drivers for AMOC and referenced Brierely and Federov (2016), who looked at model responses to the closure of the CAS to shallow waters, the opening of the Bering Strait and the closure of a deep channel between New Guinea and the equator. All three of these tectonic events occurred after 5 Ma and are therefore not relevant for the data presented here. Reviewer #3 also asked about the impact of the East-Tethys closure. As the closure occurred prior to 14 Ma we decided not to consider it in our discussion.

My main worry relates to the Poore et al (2006) data shown in Fig. 4. Reviewer #3 stated (as their 1st comment) that it did not look like the data shown by Poore et al in either Fig 9 or Fig 10. Whilst this has been edited since, it still does not appear to a direct replicate of that data. The black line could be correct, but the error bars descending to 0% and having a peak upto 100% at 9Ma are visibly wrong. I feel that the authors have dealt with the rest of the reviewer's comment (by including the Lear et al data).

We have had further correspondence with R. Samworth (co-author of Poore et al. (2006)) who advised us that the estimated error in the file he originally sent us was 2 stdev, rather than the 1 sigma that is plotted in Figure 9 of Poore et al. (2006), hence the discrepancy. For clarity and for consistency with how the Lear et al (2003) curve is plotted, we have now omitted the error frames in the updated Figure 4.

I had to look up what Northern Component Water was. The manuscript would benefit from a statement that is the palaeo-name for NADW.

A statement is now included in the discussion.

Chris Brierley (UCL Geography)

References included in the reply to reviewers

Brierley, CM, and AV Federov. Comparing the impacts of Miocene-Pliocene changes in inter-ocean gateways on climate: Central American Seaway, Bering Strait, and Indonesia. *Earth and Planetary Science Letters* **444**, 116-130 (2016).

Lear CH, Rosenthal Y, Wright JD. The closing of a seaway: ocean water masses and global climate change. *Earth and Planetary Science Letters* **210**, 425-436 (2003).

Poore HR, Samworth R, White NJ, Jones SM, McCave IN. Neogene overflow of Northern Component Water at the Greenland-Scotland Ridge. *Geochemistry Geophysics Geosystems* **7**, (2006).